# Multivariate Dense Retrieval: A Reproducibility Study under a Memory-limited Setup

**Georgios Sidiropoulos**[*]
*g.sidiropoulos@uva.nl*
*IRLab, University of Amsterdam, Amsterdam, The Netherlands*

**Samarth Bhargav**[*]
*s.bhargav@uva.nl*
*IRLab, University of Amsterdam, Amsterdam, The Netherlands*

**Panagiotis Eustratiadis**
*p.efstratiadis@uva.nl*
*IRLab, University of Amsterdam, Amsterdam, The Netherlands*

**Evangelos Kanoulas**
*e.kanoulas@uva.nl*
*IRLab, University of Amsterdam, Amsterdam, The Netherlands*

**Reviewed on OpenReview:** *https://openreview.net/forum?id=wF3ZtSlOcT*

## Abstract

The current paradigm in dense retrieval is to represent queries and passages as low-dimensional real-valued vectors using neural language models, and then compute query-passage similarity as the dot product of these vector representations. A limitation of this approach is that these learned representations cannot capture or express uncertainty. At the same time, information retrieval over large corpora contains several sources of uncertainty, such as misspelled or ambiguous text. Consequently, retrieval methods that incorporate uncertainty estimation are more likely to generalize well to such data distribution shifts. The multivariate representation learning (MRL) framework proposed by Zamani & Bendersky (2023) is the first method that works in the direction of modeling uncertainty in dense retrieval. This framework represents queries and passages as multivariate normal distributions and computes query-passage similarity as the negative Kullback-Leibler (KL) divergence between these distributions. Furthermore, MRL formulates KL divergence as a dot product, allowing for efficient first-stage retrieval using standard maximum inner product search.

In this paper, we attempt to reproduce MRL under memory constraints (e.g., an academic computational budget). In particular, we focus on a memory-limited, single GPU setup. We find that the original work (i) introduces a typographical/mathematical error early in the formulation of the method that propagates to the rest of the original paper's mathematical formulations, and (ii) does not fully specify certain important design choices that can strongly influence performance. In light of the aforementioned, we address the mathematical error and make some reasonable design choices when important details are unspecified. Additionally, we expand on the results from the original paper with a thorough ablation study which provides more insight into the impact of the framework's different components. While we confirm that MRL can have state-of-the-art performance, we could not reproduce the results reported in the original paper or uncover the reported trends against the baselines under a memory-limited setup that facilitates fair comparisons of MRL against its baselines. Our analysis offers insights as to why that is the case. Most importantly, our empirical results suggest that the variance definition in MRL does not consistently capture uncertainty. The source code for our reproducibility study is available at: https://github.com/samarthbhargav/multivariate_ir/.

---

[*]Equal Contribution

# 1 Introduction

Dense retrieval has become the new paradigm in first-stage retrieval, largely replacing lexical methods which cannot model semantic information as well as neural models. Dense retrievers following the dual-encoder architecture (Karpukhin et al., 2020) are popular first-stage retrievers due to their performance and scalability. This paradigm uses pre-trained neural language models to encode queries and passages as low-dimensional real-valued dense vectors, with relevance defined as their dot product. Passages are encoded offline and stored in a dense index. At query time, retrieval can be done efficiently using maximum inner product search (MIPS). However, representing queries and passages as single vectors has an important limitation that has influenced the research landscape: these representations do not model, capture, or express predictive uncertainty (Vilnis & McCallum, 2015). At the same time, there are various sources of uncertainty arising both from the data and the neural retrieval models:

**Query uncertainty.** User queries may include misspellings, ambiguity, and incomplete or inaccurate information (e.g., false memories). Furthermore, in a realistic setting, the retrieval system has minimal to no prior knowledge about the distribution of the queries, except for possibly a few assumptions (e.g., the language they are in, or common user mistakes).

**Passage uncertainty.** Passages may present similar uncertainty-inducing artifacts to queries, such as misspellings and ambiguity. Unlike queries, the retrieval model has little prior knowledge of the passage collection as a whole, i.e., only of the documents used during training.

**Relevance uncertainty.** Relevance, or ranking uncertainty refers to the confidence of the model in the estimated query-passage relevance. Such an estimator may be anything between a deterministic function of query and passage uncertainty, e.g., the model reproduced in this paper, to a stochastic function of deterministic query and passage representations, e.g., a Monte-Carlo dropout Bayesian estimator (Cohen et al., 2021).

Uncertainty estimation remains largely unexplored for the case of first-stage dense retrieval, despite it having received increased attention from the community in the case of re-ranking (Wang & Zhu, 2009; Zhu et al., 2009; Feng et al., 2020; Cohen et al., 2021; Heuss et al., 2023). Recently, Zamani & Bendersky (2023) proposed the multivariate representation learning (MRL) framework, the first approach that models uncertainty in the context of dense retrieval. MRL uses predictive variance as a proxy for uncertainty. Each query and passage is mapped to a multivariate Gaussian distribution parameterized by a mean vector and a (diagonal) covariance matrix, where the mean represents the predicted query or passage embedding, and the variance represents the uncertainty of said embedding. However, different from existing approaches to modelling uncertainty in IR that use Bayesian inference (Cohen et al., 2021), variance in the MRL framework does not express statistical variance, i.e., deviation from the mean, as much as it expresses predicted risk. In essence, this is a trade-off between being theoretically principled and computationally efficient. Admittedly, computational efficiency is of the utmost importance in first-stage retrieval, where one has to manage collections of potentially billions of passages.

Having represented queries and passages as multivariate normal distributions, the authors of the original paper proceed to formulate a query-passage relevance scoring function based on a simplified version of the Kullback–Leibler (KL) divergence. Further, they express this function as a dot product between query and passage representations, thereby allowing for efficient retrieval by means of standard MIPS (e.g., FAISS (Johnson et al., 2021)). Finally, they report state-of-the-art retrieval performance and show that the predicted covariance matrix can be used as a pre-retrieval query performance predictor.

Even though the results reported in the original study showcase the effectiveness of the proposed method, our study is motivated by several important questions that still need to be explored. Many of these questions have the character of a reproducibility question. First and foremost, the unavailability of the source code and model checkpoints makes it difficult to verify the paper's substantial claims. Second, in the original work, a batch of 512 was used to train MRL, indicating that substantial computational resources were available – in contrast, some of the baselines (e.g., CLDRD) were trained with a batch size of 8 on a single GPU. Third, even though the model consists of various components and several stages of knowledge distillation, the original work does not include an extensive ablation study that explores the impact of each component

on downstream performance. Therefore, it is unclear to what degree the performance gains of the proposed method stem from the multivariate representations of queries and passages. To this extent, it is important to understand the representations learned by MRL. In this reproducibility work, we aim to answer the following research questions:

**RQ1** To what extent can the results and findings of the original paper be reproduced under a memory-limited setup?

**RQ2** Can the multivariate query and passage representations express uncertainty?

**RQ3** What is the contribution of each MRL component to the downstream retrieval performance?

**RQ4** What is the impact of batch size in training MRL?

Furthermore, we summarize our contributions to the original work:

**Correction of a typographical/mathematical error.** We correct a mathematical error in the original work, made early in the formulation of the method, in an attempt by the authors to simplify the computation of KL divergence between two multivariate normal distributions. This error propagated to the rest of the original paper's mathematical formulations. We suspect that this error is typographical and that it did not leak into the technical implementation of the original authors' experiments. In fact, we provide experimental evidence that if the incorrect similarity scoring function is used instead of our corrected version, it harms retrieval performance.

**Reproducing retrieval and QPP experiments.** We reproduce the experimental setup of the original paper, for the tasks of dense retrieval and pre-retrieval query performance prediction (QPP). The original MRL model is trained with a batch size of 512, while its primary competing approach uses a batch size of 8. We explore to what extent the original work's findings are reproducible when MRL is trained with a significantly smaller batch size under a memory-limited setup that facilitates fair comparisons of MRL against its baselines (i.e., both MRL and its baselines are trained with the same batch size). We showcase that even though MRL can still achieve state-of-the-art retrieval results, retrieval performance significantly drops under a memory-limited setup. Moreover, we show that MRL does not outperform the baselines in a fair comparison. Finally, MRL yields inconsistent results for the QPP experiments, with our analysis revealing that the variance vectors do not consistently capture notions of uncertainty.

**Ablation study.** The MRL framework is composed of multiple components, including multivariate representations, knowledge distillation, and model initialization from an already effective pre-trained dense retriever. We conduct a thorough ablation study on the model's components to unveil their importance in training an effective retriever. We find that multivariate representations do not boost the model's performance, and the high effectiveness stems from the model initialization and knowledge distillation from the re-ranker. We further conduct a thorough ablation study on the effect of different batch sizes on the retrieval performance of MRL. We believe that the impressive results reported in the original paper are due to training with a large batch size for many training steps.

**Proposed improvements upon the original MRL model.** We propose a simple alteration to the original model, which results in a reduced hyperparameter search space. In short, instead of a parametric softplus activation which ensures positive semi-definiteness of the covariance matrix, we propose predicting the log-variance instead, which obviates searching for the $\beta$ hyperparameter of the softplus function. We show that the log-variance model either matches or outperforms the original softplus model.

## 2 Related Work

**Uncertainty-aware retrieval.** Uncertainty estimation in neural information retrieval (IR) has been explored in the past, although not in the context of dense (first-stage) passage retrieval, which is the main novelty aspect of the MRL method. The work of Cohen et al. (2021) and Heuss et al. (2023) focuses on risk-aware (second-stage) re-ranking. Both approaches attempt to approximate Bayesian models that predict

a distribution of relevance scores rather than point estimates; the former uses Monte-Carlo dropout (Gal & Ghahramani, 2016), while the latter leverages Laplace approximation. The common denominator across these Bayesian methods is that the predictive distribution $p(y|\theta, \mathcal{D})$ is approximated by performing forward inference using multiple samples of $\theta$. This is the main difference between prior work and MRL: In MRL, predictive uncertainty is not framed as weight uncertainty, and variance does not represent deviation from the mean prediction. Rather, variance in MRL is a predicted value of a deterministic estimator.

**Uncertainty for detecting out-of-distribution/corruptions.** While a variety of efforts exist in the area of stochastic representations in image retrieval (Warburg et al., 2021; Chun et al., 2021), recent work by Warburg et al. (2023) showed that Bayesian image retrieval with Laplace approximation can achieve some desirable properties. They show that the uncertainty of prediction increases (almost monotonically) with the amount of corruption in the input. The model's predictive uncertainty further behaves as expected when making out-of-domain predictions. These insights are valuable in text retrieval as well, where these desirable properties have not yet been achieved effectively.

**Knowledge distillation.** The MRL framework is surrounded by multiple layers of knowledge/parameter distillation, which we summarize in this section. First, the selected neural architecture employed in MRL is DistilBERT (Sanh et al., 2019); a distilled version of BERT with 40% fewer parameters for 97% of its original performance. Furthermore, the architecture has been distilled with balanced topic-aware sampling (TAS-B) (Hofstätter et al., 2021), which uses two teacher models to construct better training batches. Finally, MRL itself uses a knowledge distillation loss inspired by CLDRD (Zeng et al., 2022). While the original paper does not discuss how these sources of distilled knowledge affect downstream performance, in Section 4 of this paper we perform a thorough ablation study that examines them one-by-one.

## 3 Methodology

The proposed MRL framework represents queries and passages as multivariate Gaussian distributions. It does so by computing a mean vector $\mu$ and a diagonal covariance matrix $\Sigma$ for a query $q$ and passage $d$, using query and passage encoders $f_\theta$ and $f_\phi$, parameterized by $\theta$ and $\phi$ respectively,

$$(\mu_Q, \Sigma_Q) = f_\theta(q), \tag{1}$$

$$(\mu_D, \Sigma_D) = f_\phi(d). \tag{2}$$

It is also possible to have $\theta = \phi$, i.e., weight sharing, which the authors of the original paper opt for. Dense retrieval models (e.g., DPR, Karpukhin et al., 2020) typically utilize the embedding of a special token, the `[CLS]` token as the low-dimensional representation of queries and documents. Given a piece of text, for instance, "Hello world", pre-processing appends the special `[CLS]` token to the start of the text, producing "`[CLS]` Hello world", and the output of the transformer model for the `[CLS]` token is used. Relevance is a function of query and document representations, typically a dot product or cosine similarity. MRL, however, produces two vectors per input, which motivates the choice in the original study to use an additional special token, termed the `[VAR]` token, appended after the `[CLS]` token, but before the text. For instance, "Hello world" is pre-processed to "`[CLS]` `[VAR]` Hello world". The output representation of the `[CLS]` token is used to compute the mean, and the output representation of the `[VAR]` token is used to compute the variance.

The relevance score between queries and passages is then defined as the negative KL divergence between their distributional embeddings: $Q \sim \mathcal{N}(\mu_Q, \Sigma_Q)$ and $D \sim \mathcal{N}(\mu_D, \Sigma_D)$,

$$\text{rel}(q, d) = -\text{KLD}(Q\|D). \tag{3}$$

The minus sign is there to implement a "higher is better" type of scoring. To simplify matters, we will disregard it in the upcoming derivations and re-introduce it at the very end. In this section, we detail the reproducibility study of the above framework. First, we direct attention to a small mathematical error that was made early in the formulation of the relevance scoring in the original paper, that propagated through the rest of the mathematical derivations. We then discuss matters of model training. Whenever we make a strong assumption due to the lack of implementation detail in the original paper, or the lack of shared source code, it is explicitly mentioned.

### 3.1 KL divergence-based relevance scoring

In Eq. 4, we start by repeating the standard definition of KL divergence, as written in Eq. 9 of the original paper:

$$\text{KLD}(Q\|D) = \frac{1}{2}\Big[\log\frac{\det\Sigma_D}{\det\Sigma_Q} - k + \text{tr}\{\Sigma_D^{-1}\Sigma_Q\} + (\mu_Q - \mu_D)^\intercal \Sigma_D^{-1}(\mu_Q - \mu_D)\Big], \tag{4}$$

where $k$ denotes the dimensionality of the multivariate Gaussian embeddings. For the purpose of relevance scoring, the authors proceed to further simplify Eq. 4 and reformulate it as a document ranking function. They do so by eliminating document-independent terms and constants, and by taking advantage of the fact that the covariance matrices are diagonal. Let us follow their simplification steps by considering each term separately. For the first term we have,

$$\log\frac{\det\Sigma_D}{\det\Sigma_Q} = \log\det\Sigma_D - \underbrace{\log\det\Sigma_Q}_{\substack{\text{constant w.r.t.}\\\text{doc. ranking}}} = \log\det\Sigma_D = \log\prod_{i=1}^{k}\sigma_{i_D}^2 = \sum_{i=1}^{k}\log\sigma_{i_D}^2. \tag{5}$$

The subsequent steps in the original paper contain an error in the simplification of the second term. We include the original formulation in Appendix A for completeness. We note that using the original formulation leads to drastically lower performance, which makes it likely that this error is typographical i.e., it did not propagate to the implementation (see Section 5.1 for more details). We provide the correct derivation in Eq. 6 as follows:

$$\text{tr}\{\Sigma_D^{-1}\Sigma_Q\} = \sum_{i=1}^{k}\frac{\sigma_{i_Q}^2}{\sigma_{i_D}^2}. \tag{6}$$

Finally, for the third term we have,

$$(\mu_Q - \mu_D)^\intercal \Sigma_D^{-1}(\mu_Q - \mu_D) = \sum_{i=1}^{k}\frac{(\mu_{i_Q} - \mu_{i_D})^2}{\sigma_{i_D}^2} = \sum_{i=1}^{k}\frac{\mu_{i_Q}^2}{\sigma_{i_D}^2} - \sum_{i=1}^{k}\frac{2\mu_{i_Q}\mu_{i_D}}{\sigma_{i_D}^2} + \sum_{i=1}^{k}\frac{\mu_{i_D}^2}{\sigma_{i_D}^2}. \tag{7}$$

Combining Eq. 5, 6 and 7 into Eq. 4, and removing constants, we arrive at the intended derivation of the ranking function:

$$\text{KLD}(Q\|D) = \sum_{i=1}^{k}\log\sigma_{i_D}^2 + \sum_{i=1}^{k}\frac{\sigma_{i_Q}^2}{\sigma_{i_D}^2} + \sum_{i=1}^{k}\frac{\mu_{i_Q}^2}{\sigma_{i_D}^2} - \sum_{i=1}^{k}\frac{2\mu_{i_Q}\mu_{i_D}}{\sigma_{i_D}^2} + \sum_{i=1}^{k}\frac{\mu_{i_D}^2}{\sigma_{i_D}^2}. \tag{8}$$

Note that unlike Eq. 4, Eq. 8 is no longer the KL divergence. After all the simplifications, it is a KL divergence-based relevance scoring function for *ranking* documents given a query. From this point forward, we continue with the work described in the original paper, but we base it on our Eq. 8, which is the derivation of the relevance scoring function that includes our correction.

The next step of this reproducibility study is to express Eq. 8 as a dot product between query and passage vectors i.e., $\text{KLD}(Q\|D) = q^\intercal \cdot d$, with the purpose of reusing standard efficient inner product similarity search (Johnson et al., 2021). To do so, we isolate the document-specific terms of Eq. 8 that can be precomputed:

$$\gamma_D = \sum_{i=1}^{k}\left(\log\sigma_{i_D}^2 + \frac{\mu_{i_D}^2}{\sigma_{i_D}^2}\right). \tag{9}$$

In the original paper, the term $\gamma_D$ is referred to as a "document prior". Now we can express the relevance score as a dot product between query and passage vector representations:

$$\vec{q} = \left[1, \sigma_{1_Q}^2, \dots, \sigma_{k_Q}^2, \mu_{1_Q}^2, \dots, \mu_{k_Q}^2, \mu_{1_Q}, \dots, \mu_{k_Q}\right], \tag{10}$$

$$\vec{d} = \left[\gamma_D, \frac{1}{\sigma_{1_D}^2}, \dots, \frac{1}{\sigma_{k_D}^2}, \frac{1}{\sigma_{1_D}^2}, \dots, \frac{1}{\sigma_{k_D}^2}, -\frac{2\mu_{1_D}}{\sigma_{1_D}^2}, \dots, -\frac{2\mu_{k_D}}{\sigma_{k_D}^2}\right], \tag{11}$$

where $\vec{q}, \vec{d} \in \mathbb{R}^{1\times(3k+1)}$. At this point, we remind the reader that following Eq. 3, the relevance score is the negative of the KL divergence.

## 3.2 Listwise knowledge distillation

Knowledge distillation has become of great importance in boosting the effectiveness of dense retrievers (Hofstätter et al., 2020). In detail, a highly effective cross-encoder re-ranker is used as a teacher to transfer knowledge to a less effective but efficient first-stage dense retriever student model. Consequently, the effectiveness of the dense retriever is increased while it retains its efficiency. In the original work by Zamani & Bendersky (2023), the authors employ a listwise distillation loss function (Zeng et al., 2022) to train their dense retriever (i.e., student model). For each query $q$ and a set of passages $D_q$ (later in this section we provide details on how this set is constructed), the loss is computed as:

$$\sum_{d,d' \in D_q} \mathbb{1}\{y_q^t(d) > y_q^t(d')\} \left| \frac{1}{\pi_q(d)} - \frac{1}{\pi_q(d')} \right| \log(1 + e^{M_\theta(q,d') - M_\theta(q,d)}), \tag{12}$$

where $\pi_q(d)$ denotes the position of passage $d$ in the ranked list produced by the dense retrieval student model $M_\theta$ and $y_q^t(d)$ denotes the relevance judgment produced by the teacher model for the pair of query $q$ and passage $d$; $y_q^t(d)$ can be either a score or a label. In the original work of Zamani & Bendersky (2023), $y_q^t(d)$ is the raw score from the teacher model for a query-passage pair, and $D_q$ is constructed as follows. Given a query $q$, the passage set $D_q$ is constructed with positive passages provided by the dataset's official relevance judgments. On the other hand, the negative passages are sampled from the top-k passages retrieved with BM25 and the top-k passages retrieved by the student dense retrieval model itself. Finally, the original work uses in-batch negative training to reuse passages from other queries that are already in the batch.

# 4 Experimental Setup

## 4.1 Datasets and metrics

Our evaluation is performed on both in-domain (ID) and out-of-domain (OOD) data. In the OOD setting, we perform zero-shot evaluation. All tasks are ad-hoc retrieval, with a fixed set of documents. Statistics of the datasets are reported in Appendix D. We summarize the datasets and evaluation methodology below.

**In Domain (ID).** We train all models on the MS-MARCO (Nguyen et al., 2016) training set. Note that we split the full training set into a training and validation set for hyperparameter tuning as described in Section 4.4. There are three in-domain evaluation sets, all of which are based on the MS-MARCO corpus. This includes the MS-MARCO Dev set, the TREC-DL 2019 (Craswell et al., 2020) and TREC-DL 2020 (Craswell et al., 2021) datasets. Both TREC datasets are densely labeled by humans. The evaluation metric for the Dev set is the mean reciprocal rank (MRR) with a cut-off of 10, denoted as MRR@10. For the TREC subsets, we use the standard evaluation metrics of normalized discounted cumulative gain at 10 (nDCG@10), and mean average precision (MAP).

**Out of Domain (OOD).** We evaluate the retrieval models' generalization ability in different domains via zero-shot passage retrieval experimentation. All retrieval models are trained on the MS-MARCO training set and tested on previously unseen queries and underlying corpus. We replicate the evaluation setup outlined in Zamani & Bendersky (2023), with nDCG@10 as the primary metric. We evaluate the following OOD datasets in zero-shot setting: (i) SciFact (Wadden et al., 2020): a scientific claim verification dataset where the task involves retrieving abstracts that either refute or support a claim, (ii) FiQA (Maia et al., 2018): a dataset that involves retrieval of documents in the financial domain using natural language questions, (iii) TREC-COVID (Voorhees et al., 2021): a biomedical dataset of scientific articles about COVID-19, with questions as the topics/queries, and (iv) CQADupStack (Hoogeveen et al., 2015): a community question answering (CQA) dataset, with the task of retrieving duplicate questions in a community website (StackOverflow).

## 4.2 Baselines

We compare MRL against the following single-vector dense retrieval models:

- **DPR** (Karpukhin et al., 2020): is a traditional dense retriever that is trained with softmax cross-entropy.

- **TAS-B** (Hofstätter et al., 2021): is an effective dense retriever that is trained by combining (i) knowledge distillation from a re-ranker teacher model (i.e., cross-encoder) with (ii) a balanced topic-aware sampling method. This method alternates the creation process of the training batches by composing batches based on queries clustered in the same topic. Furthermore, it selects passage w.r.t. the pairwise margin between positive and negative passages in the batch so that the margin of positive-negative pairs is balanced in the margin range.

- **CLDRD** (Zeng et al., 2022): is a state-of-the-art dense retriever that uses TAS-B as initialization and is trained by combining curriculum learning with knowledge distillation; in particular, it uses the listwise loss of Eq. 12. The student dense retriever is trained via an iterative training process in which the difficulty of the training data, produced by the re-ranking teacher model, increases with each iteration. Additional information regarding the training setup of CLDRD can be found in the Appendix C.

The motivation behind selecting these baselines is twofold: First, their inclusion in the original study, and second, to enable fair comparisons in our subsequent ablation study. For instance, MRL can be compared with CLDRD to assess the impact of the multivariate representations, and a similar assessment can be made when MRL without distillation is compared against DPR.

### 4.3   Query performance prediction

The QPP task (He & Ounis, 2004; 2006; Carmel & Yom-Tov, 2010) involves inferring the difficulty of a given query for a search system without using relevance judgments. We replicate the pre-retrieval QPP setup in Zamani & Bendersky (2023), evaluating on the TREC-DL'19 and TREC-DL'20 datasets. That is, we retrieve documents for a given query using a search system and evaluate it using nDCG@10 to obtain a ground-truth assessment of performance. Then, we use a QPP method to predict the performance and evaluate it against the ground-truth assessment using three correlation measures, Spearman's correlation, Pearson correlation, and Kendall's Tau.

The effectiveness of a QPP method is a function of the underlying retrieval system. Since it was unclear from the original study which system was used to compute the ground truth performance, we experiment with multiple search systems.

In addition to the model itself, we experiment with three retrieval models independent of the MRL model, to measure how well the QPP method generalizes. We include a traditional lexical retriever (BM25), a simple dense retriever (DPR), and an effective dense retriever (TAS-B). We use the following baselines used in the original study:

- **SCQ** (Zhao et al., 2008): computes the similarity between a query and the corpus for each query term based on the frequency of occurrence of the term in the corpus.

- **VAR** (Carmel & Yom-Tov, 2010): considers the variance or standard-deviation of the term weights of each query term, based on the documents in which the term occurs.

- **IDF** (Carmel & Yom-Tov, 2010): is based on the inverse document frequency of each query term.

- **PMI** (Hauff, 2010): is a predictor that computes the pointwise mutual information, assigning high scores for frequently co-occurring query terms. Given all possible query term pairs, either the average or the maximum can be used as the predictor.

For SCQ, VAR, and IDF, the scores are computed at the query term level and then aggregated using either summing, averaging, or taking the maximum of each score. We report each of these aggregations in the results.

**QPP for MRL.** Zamani & Bendersky (2023) mention that the norm of the variance $|\Sigma_Q|$ is used to compute the predicted performance. We interpret this statement as using a *function* of the covariance, and use the negative norm $-|\Sigma_Q|$, because the predicted variance should *increase* for queries that are difficult, rather than decrease. For instance, a typographical error in the query makes it more difficult to address than a

"clean" query (Sidiropoulos & Kanoulas, 2022; 2024), leading to lower performance compared to a clean query. Similarly, an OOD query could also result in poorer performance on average compared to an ID query. From the QPP perspective, a model should therefore assign *lower* predicted performance for these types of queries, motivating our choice. In Section 5.2.1, we show that this intuition holds empirically.

### 4.4 Implementation details

We train MRL for $200K$ steps. In each step, we optimize the distillation loss (Eq. 12) using a batch of queries, one positive passage per query, and 30 negative passages per query; 5 of the negative passages are mined with BM25, and 25 are mined with the student model. With that setup we use a batch size of 15 queries – the maximum that can fit in a 40GB A100 GPU, given the size of our model. We set the maximum length for queries and passages to 32 and 256 tokens, respectively. We initialize the dense retriever student model with the official TAS-B checkpoint, and we set as the teacher model the `ms-marco-MiniLM-L-6-v2` cross-encoder that is publicly available on HuggingFace. We use an Adam optimizer with a learning rate of $5 \times 10^{-6}$, and linear learning rate scheduling with warm-up for 10% of the training steps. The $\beta$ parameter for softplus is set to 2.5. For MRL, the mean and variance are obtained by passing the `[CLS]` token and a `[VAR]` token respectively through fully connected projection layers. The MRL models reported use means and variances projected down to 383 ($= \frac{768}{2} - 1$).

Since MS-MARCO does not include a validation set, we split the training set into a validation (6890 queries) and a training set. The parameters above were selected after a hyperparameter search with the validation set performance used to pick the best model. Refer to Appendix E for the full set of hyperparameters. We use the `Tevatron` toolkit (Gao et al., 2023) to train the models and the `pytrec_eval` library (Van Gysel & de Rijke, 2018) to evaluate the retrieval performance. Finally, our QPP baselines are based on an existing implementation by Meng et al. (2023).

## 5 Discussion

We organize the discussion section around retrieval experiments in Section 5.1, the investigation of the variance vectors in Section 5.2, and the results of the ablation study in Section 5.3.

### 5.1 Reproducing the retrieval results

In this study, we work under a memory-limited setup – an academic computational budget with constrained access to a single GPU. We start by testing whether we can obtain the results reported in the original study under these memory constraints. We report the results in Table 1, where DPR, CLDRD, and MRL are our implementations of the original methods. We report results for TAS-B by using the official pre-trained checkpoint, which also serves as CLDRD and MRL initialization. This way, we can ensure a fair comparison between the different methods. We include the original numbers in the lower group in Table 1. At this point, we want to underline that for MRL we use the corrected KL formulation we presented in Section 3 for our experiments. Our decision to do so is grounded in the belief that the original study's authors also used this formulation in their implementation and that the formulation with the mathematical error is a typographical mistake in their paper. We arrived at this conclusion based on our preliminary experiments, which yielded a dramatically low retrieval performance (i.e., MRR@10 was 0.229 for MS-MARCO) when following the wrong formulation.

We first focus on testing whether we are able to replicate the results for CLDRD, the main competitor of MRL. As shown in Table 1, our experimental results affirm the state-of-the-art retrieval performance (for single-vector dense retrievers) of CLDRD. Furthermore, our findings validate the original study regarding its ability to enhance the performance of TAS-B, both in the ID and OOD scenarios. We consider this a successful replication despite the slight discrepancy in the results. The reason for not obtaining the exact same results can be attributed to different development toolkits, hardware, implementation details not present in the original work, etc.

Table 1: Reproduction results of MRL on our memory-limited setup. The *Reproduced* section contains the reproduction results, while the *Reported* section contains the results reported in the original study. The lower half of the table reports the results of contemporary dense retrieval methods. Note: MRL remains, to date, the only work with uncertainty estimation capabilities in dense retrieval, and the methods reported in the lower half are simply for reference.

| | Model | MS MARCO | | TREC-DL'19 | | TREC-DL'20 | | SciFact | FiQA | TREC-COVID | CQADupStack |
|---|---|---|---|---|---|---|---|---|---|---|---|
| | | MRR@10 | MAP | NDCG@10 | MAP | NDCG@10 | MAP | NDCG@10 | NDCG@10 | NDCG@10 | NDCG@10 |
| Reproduced | DPR | .312 | .319 | .649 | .345 | .625 | .356 | .474 | .231 | .600 | .266 |
| | TAS-B | .344 | .351 | .721 | .396 | **.685** | .430 | **.643** | .301 | .481 | .313 |
| | CLDRD | **.378** | **.383** | **.727** | .448 | .670 | .446 | .627 | **.308** | **.608** | **.327** |
| | MRL | .373 | .378 | .707 | **.462** | .681 | **.461** | .591 | .291 | .497 | **.327** |
| Reported | TAS-B | .344 | .351 | .717 | .447 | .685 | .455 | .643 | .300 | .481 | .314 |
| | CLDRD | .382 | .386 | .725 | .453 | .687 | .465 | .637 | .348 | .571 | .327 |
| | MRL | .393 | .402 | .738 | .472 | .701 | .479 | .683 | .371 | .668 | .341 |
| **Recent Work in Dense Retrieval (For Reference)** | | | | | | | | | | | |
| | SimLM Wang et al. (2023) | .411 | - | .714 | - | .697 | - | - | - | - | - |
| | ADAM Tao et al. (2024) | .41 | - | .734 | - | - | - | .594 | .315 | .730 | - |
| | Llama2Vec Li et al. (2024) | .431 | - | .734 | - | .729 | - | .748 | .485 | .869 | .432 |

Regarding the reproduction of MRL, as shown in Table 1, we were unable to obtain the same results as the original study. We observe a drop in performance across all reported metrics for both the ID and OOD datasets. For instance, in the case of FiQA, our implementation yielded an NDCG@10 of 0.291, which is lower than the original study's reported value of 0.371. Furthermore, we could not find the trends that were reported in the original study. Specifically, the original study showed that MRL outperformed both TAS-B and CLDRD in ID and OOD scenarios. In contrast, in our work, MRL achieves similar performance to CLDRD in the ID datasets. A similar trend holds for the OOD dataset, except for TREC-COVID, where CLDRD outperforms MRL with a substantially higher score of 0.608 compared to 0.497 for MRL. Upon comparing MRL with TAS-B, we have mixed results, with MRL matching or outperforming TAS-B for some datasets and metrics, but faring worse for others. For the OOD datasets, MRL surpasses TAS-B only for TREC-COVID and CQADupStack.

We stress that our goal in this work is not to replicate MRL; an exact replication is not possible since the original manuscript does not specify several design choices – the best hyperparameters (i.e., learning rate, *softplus* $\beta$, number of training steps), the cross-encoder model that was used as a teacher, as well as the number of negative passages per query in a batch. We aim to reproduce MRL under our memory-limited setup, which facilitates fair comparisons against the baselines. From our experimental results, we conclude that although MRL is a competitive approach in our setup, the multivariate representations do not boost the retrieval performance. Furthermore, we unveil that MRL cannot consistently outperform its competitors when evaluated under fair comparisons. However, different from CLDRD, MRL produces a variance that can be used in downstream tasks. We investigate the utility of this predicted variance in the following section.

## 5.2 Analyzing the variance

We analyze the predicted variance in three experiments: query performance prediction experiments (Section 5.2.1), experiments with typos, and retrieval experiments with alternative encoding schemes in Section 5.2.2. The first is a replication of the QPP experiments included in the original paper, while the latter two are additional analyses.

### 5.2.1 Query performance prediction

The results for the QPP experiments are plotted in Figures 1 and 2. We also report the MRL results separately in Table 2, which also includes the reported numbers in Zamani & Bendersky (2023). As mentioned previously, we used four reference models because the original study did not report which model was used, and also to see if MRL generalizes to different reference models. We note that the original paper does not mention how the norm is used in computing the predicted performance – in our experiments, we use the

Table 2: QPP results for MRL for four reference models. While MRL does perform well for TREC-DL'20 for BM25, it fails to do so for TREC-DL'19. The opposite is true for DPR. For both TAS-B and MRL reference models, MRL is more consistent but fails to reach the reported performance (bottom row). Furthermore, as is evident from Figures 1 and 2, MRL is outperformed by simple baselines in most comparisons.

| | | TREC-DL 19 | | | TREC-DL 20 | | |
|---|---|---|---|---|---|---|---|
| | | S-$\rho$ | P-$\rho$ | K-$\tau$ | S-$\rho$ | P-$\rho$ | K-$\tau$ |
| MRL | BM25 | -.029 | -.034 | -.014 | .275 | .296 | .190 |
| | DPR | .233 | .171 | .155 | -.051 | -.013 | -.041 |
| | TAS-B | .204 | .196 | .138 | .177 | .166 | .117 |
| | MRL | .171 | .182 | .134 | .157 | .185 | .106 |
| | MRL (reported) | - | .271 | .259 | - | .272 | .298 |

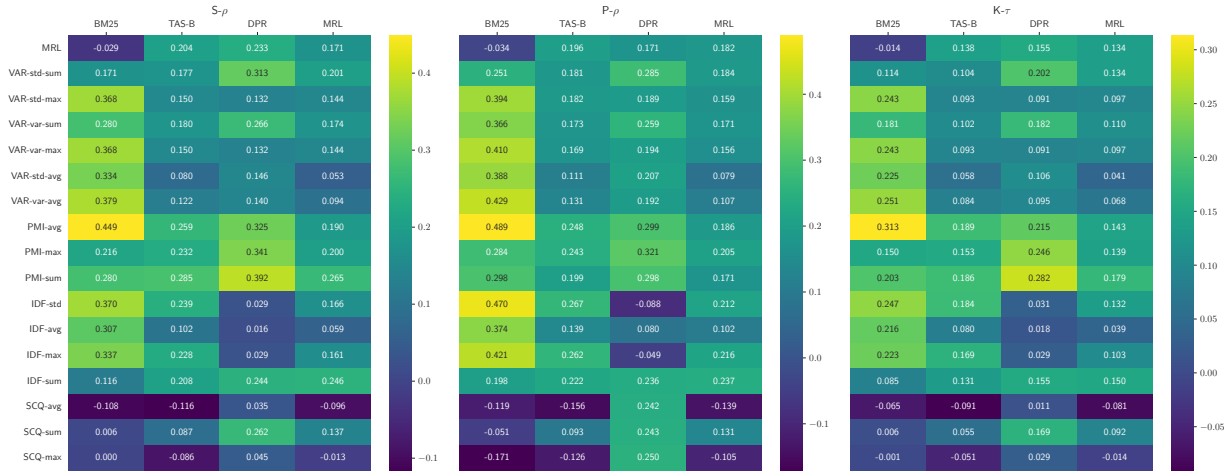

Figure 1: QPP results for TREC-DL'19 for four reference models (x-axis) and several methods (y-axis). Each subplot corresponds to a different correlation metric: Spearman's (S-$\rho$), Pearson's (P-$\rho$), and Kendall's Tau (K-$\tau$) correlations. MRL is outperformed by simple baselines regardless of the reference model or metric.

*negative* norm (using the norm flips the signs of the correlation) – intuitively, a higher uncertainty should result in lower performance (see Section 4.3).

From Table 2, we were unable to reproduce the numbers for MRL reported in the original paper, with any of the reference models. While MRL achieves higher than reported numbers for TREC-DL'20 with BM25, this result does not hold for TREC-DL'19, where there appears to be a random correlation. This trend is flipped with DPR as the reference – MRL performs well for TREC-DL'19 but not TREC-DL'20. For TAS-B and DPR, MRL is more consistent. However, the correlation obtained is lower than in the original study. But, how does MRL compare with the baselines?

Figure 1 contains results for TREC-DL'19, with each subplot corresponding to the three metrics we used. Comparing MRL (top row) with the other methods, we notice that at least one baseline outperforms MRL for each metric regardless of the reference model. In particular, at least one variant of the PMI baseline outperforms MRL. We remind the reader that these baselines are simple, non-parametric methods that use statistics derived from the corpora to compute the query difficulty.

For TREC-DL'20, the results are more encouraging. If the reference model is MRL itself, we observe that MRL beats every other baseline for all three metrics. However, this result does not generalize to the other three reference models, where similar trends to TREC-DL'19 are observed, with at least one baseline (among

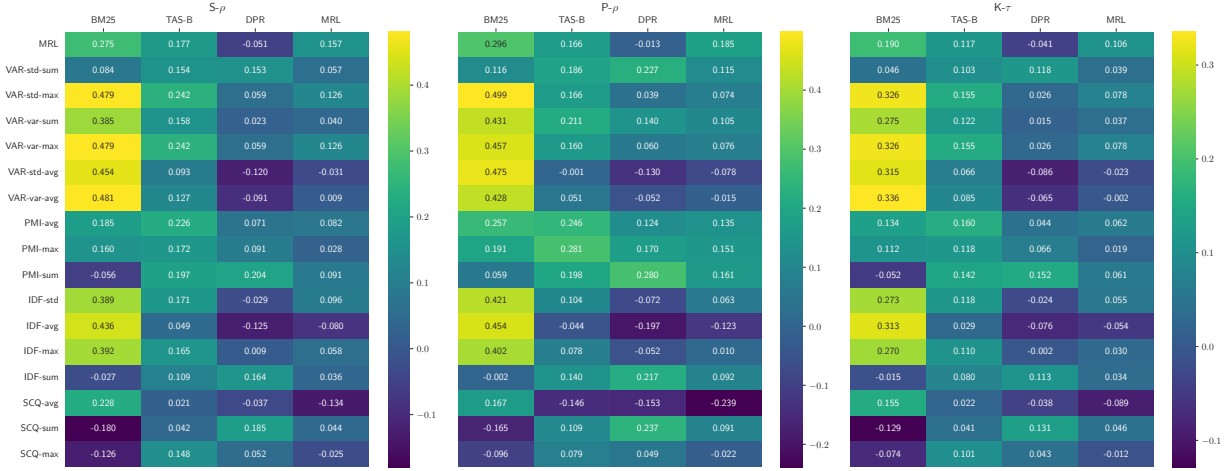

Figure 2: QPP results for TREC-DL'20. Simple lexical baselines outperform MRL when the reference is BM25, TAS-B, or DPR. With MRL itself as the reference, MRL achieves better performance, often by large margins.

PMI, VAR, IDF) beating MRL. We observe similar trends for three reference models BM25, TAS-B, and DPR, with MRL outperformed by at least one baseline.

In these experiments, we investigate the degree to which the MRL framework captures the notion of uncertainty by measuring a proxy – query difficulty. Intuitively, this suggests that for some datasets and reference models, higher uncertainty was indeed assigned to difficult queries. However, the positive correlation is *not consistent* across datasets. MRL fails to generalize to different reference models and datasets, achieving random correlation in many settings. Furthermore, MRL is outperformed by simple non-parametric baselines in most comparisons. The lack of consistent and strong correlations suggests that MRL is unlikely to be a strong and consistent predictor of query difficulty in our experimental setup. Motivated by these results, we explore what the predicted variance captures in the next section.

### 5.2.2 Does MRL capture uncertainty?

We outline two additional experiments investigating the predicted variance beyond the original paper: (a) contrasting the predicted variance of corrupted and clean data and (b) experimenting with alternate encoding schemes.

**Experiments with typographical errors.** Here, we consider an analog to the QPP experiments above, but instead of retrieval difficulty, we examine if the model is sensitive to *data distribution shifts* instead. Inspired by works in the vision domain (e.g., Warburg et al., 2021; 2023), we argue that a model should assign higher variance to *corrupted* or *OOD data* compared to clean or ID data.

We experiment with the DL-Typo (Zhuang & Zuccon, 2022) dataset that contains 60 query pairs accompanied by relevance assessments. Each pair consists of a real user query with typographical errors and its corresponding version where these errors have been corrected. For instance, the corrupted query "what is acid reflex" and its typo-free version "what is acid reflux". Given these data, we compute the norms of the predicted variance and plot their distributions. Suppose the model indeed models uncertainty accurately. In that case, we expect (a) clean data should be assigned lower variance compared to corrupted data, (b) the distributions of the clean and corrupted data are well separated, and (c) the differences between the clean and corrupted data, i.e., $|\Sigma_{\text{clean}}| - |\Sigma_{\text{corrupted}}|$, should be negative.

We plot these distributions in Figure 3. While we expected more variance to be assigned to the corrupted data, the opposite is true. There is some separation observed between the two distributions – the distribution of the corrupted data ("typo") is to the left of the clean data. The right plot underscores this result, as the

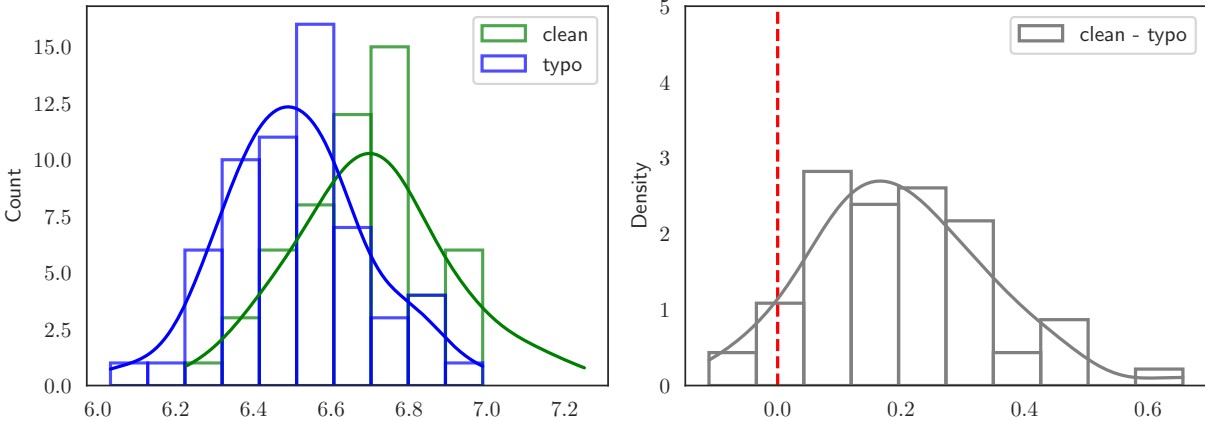

Figure 3: Uncertainty of clean and corrupted data: We plot the distributions of the norm of the predicted variance of the clean and corrupted data on the left. MRL assigns *lower* uncertainty to the corrupted data compared to the clean data. On the right, we plot the distribution of $|\Sigma_{\text{clean}}| - |\Sigma_{\text{corrupted}}|$, which is mostly positive (i.e., right of the red-dotted line). Contrary to expectations, MRL fails to assign higher uncertainty to corrupted data.

Table 3: Performance of different encoding schemes for the MRL model. The first row uses the original equations proposed in the original paper, whereas the second row includes our corrections.

| Encoding | MS MARCO | | TREC-DL'19 | | TREC-DL'20 | | SciFact | FiQA | TREC-COVID | CQADupStack |
|---|---|---|---|---|---|---|---|---|---|---|
| | MRR@10 | MAP | NDCG@10 | MAP | NDCG@10 | MAP | NDCG@10 | NDCG@10 | NDCG@10 | NDCG@10 |
| Zamani & Bendersky (2023) (Eq. 18 & 19) | .229 | .236 | .525 | .255 | .486 | .259 | .096 | .116 | .215 | .183 |
| Ours (Eq. 10 & 11) | .373 | .378 | .707 | .462 | .681 | .461 | .591 | .291 | .497 | .327 |
| Mean | .279 | .285 | .624 | .314 | .557 | .311 | .382 | .168 | .145 | .187 |

$|\Sigma_{\text{clean}}| - |\Sigma_{\text{corrupted}}|$ distribution is mostly positive. This suggests that contrary to expectation, the model predicts a higher variance for clean data. We expand on this analysis by examining the role of the predicted variance in retrieval relevance.

**Encoding.** MRL produces both a mean and variance for queries and documents. In Table 1, we reported retrieval results using the encoding scheme which enables retrieval using the KL divergence, i.e., documents are encoded and indexed with Eq. 11 and queries with Eq. 10. If the variance *only* captures uncertainty, we argue that the difference in retrieval performance when *only* the mean is used should not be much lower than the performance obtained with this encoding scheme. However, as Table 3 shows, this is not true. Comparing the encoding scheme (second row) with using just the mean (third row), we see that performance drops sharply across all datasets, especially for the OOD datasets.

The performance drop could be explained partly due to the way the models were trained. Since the full KL loss (Eq. 4) was used in training the model, it may not be equipped to perform retrieval using just the mean. However, intuitively, we expect the mean to model *relevance* and variance to model *uncertainty*, which means that the retrieval performance should not be drastically different when only the mean is used for retrieval. The drastic drop suggests that the model may be instead using the predicted variance vectors as a signal for relevance. This is the core difference between a method such as MRL and a Bayesian method: variance in MRL is not statistical variance, i.e., it does not express deviation from the mean prediction. Instead, variance in MRL is a deterministically estimated quantity that minimizes a distance objective.

In this section, we examined if the MRL model consistently predicts a notion of variance that reflects a notion of uncertainty defined by either query difficulty or sensitivity to data distribution shifts. We find that the QPP results are inconsistent, and against initial expectations, the model assigns a higher uncertainty

Table 4: Ablation study on the different components of the MRL framework. The first part of the table examines the influence of initialization (i.e., TAS-B), loss function (i.e., Listwise KD), and multivariate representations for queries and passages on MRL performance. The second part explores the impact of training MRL using the curriculum learning framework of CLDRD on performance.

| # | Variant | MS MARCO MRR@10 | TREC-DL'19 NDCG@10 | TREC-DL'20 NDCG@10 | SciFact NDCG@10 | FiQA NDCG@10 | TREC-COVID NDCG@10 | CQADupStack NDCG@10 |
|---|---------|-----------------|--------------------|--------------------|-----------------|--------------|--------------------|---------------------|
| 1 | MRL
  Multivariate representation
  TAS-B
  Listwise KD
  Teacher raw scores
  Qrels & Negative Mining | .373 | .707 | .681 | .591 | .291 | .497 | .327 |
| 2 | - Multivariate representation
+ Vector representation | .375 | .719 | .686 | .637 | .306 | .618 | .334 |
| 3 | - TAS-B
+ DistilBERT | .356 | .693 | .677 | .567 | .264 | .536 | .320 |
| 4 | - Listwise KD
+ Cross-entropy | .328 | .629 | .644 | .498 | .245 | .473 | .268 |
| 5 | MRL-CLDRD
  - Teacher raw scores
  - Qrels & negative mining
  + Teacher constructs the batch
  + Teacher pseudolabels | .375 | .721 | .667 | .605 | .293 | .510 | .320 |
| 6 | CLDRD
  - Teacher raw scores
  - Qrels & negative mining
  - Multivariate representations
  + Teacher constructs the batch
  + Teacher pseudolabels
  + Vector representations | .378 | .727 | .670 | .627 | .308 | .608 | .327 |

to corrupted data. In addition, experiments with encoding using only the mean suggest that the variance seems to model relevance since performance drops drastically when only the mean is used for retrieval.

## 5.3 Ablation study

Even though the MRL performance reported in the original paper was not reproduced under our experimental setup, MRL remains a framework that can produce a highly effective dense retriever even under memory constraints. MRL consists of several components: multivariate representations, knowledge distillation, model initialization from a pre-trained dense retriever, and a batch construction strategy. Thus, it needs to be made clear how much each component impacts effectiveness. We expand on the original paper's findings by studying each component's contribution to retrieval performance through experimentation with various MRL variants.

**Multivariate representations.** MRL represents queries and passages as multivariate distributions and uses negative multivariate KL divergence to compute similarity. First and foremost, we want to understand how much the multivariate representations contribute to the overall effectiveness of the retriever. In order to test this, we conduct an experiment where we substitute multivariate representations with vector representations and compute similarity via the dot product. We find that multivariate representations do not lead to a higher retrieval performance; in contrast, we observe a small decrease in performance when used (see rows 1 and 2 in Table 4).

**Model initialization.** Since MRL is initialized with TAS-B, a pre-trained, highly effective dense retriever, it is important to examine the impact of its initialization on its downstream performance. To test this, we use DistilBERT as the initialization instead. When comparing row 1 against row 3 in Table 4, we see that it is possible to train a competitive model with a DistilBERT initialization; however, it does not achieve state-of-the-art performance.

**Loss function.** MRL can be trained using supervised contrastive learning by minimizing a softmax cross-entropy loss (Karpukhin et al., 2020), instead of training with knowledge distillation. In row 4 of Table 4, we test this alternative and find that following a knowledge distillation training scheme is crucial for training an

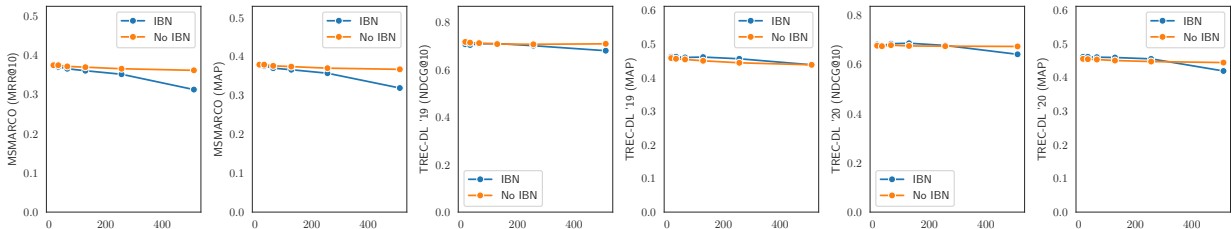

Figure 4: Impact of batch size on the MRL performance for ID datasets. IBN and No IBN indicate training MRL with in-batch and no in-batch negatives, respectively.

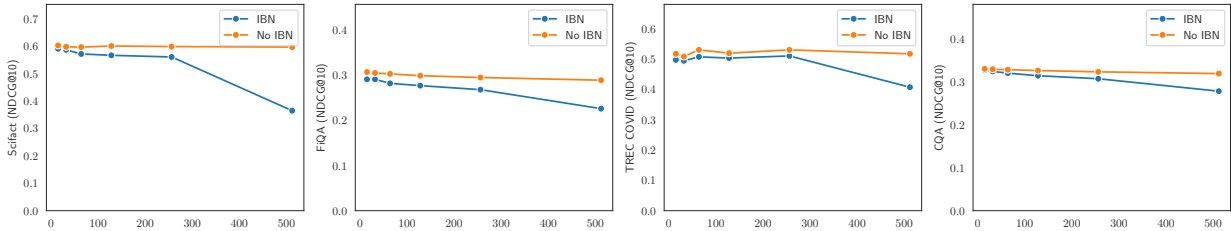

Figure 5: Impact of batch size on the MRL performance for OOD datasets. IBN and No IBN indicate training MRL with in-batch and no in-batch negatives, respectively.

effective retriever. This result is in line with recent work in dense retrieval that have shown the efficacy of knowledge distillation training over supervised contrastive learning (Hofstätter et al., 2021; Lin et al., 2021).

**Negatives from the student model.** The MRL framework uses the student model to sample hard negatives (see Section 3.2). This can be achieved by using a recent checkpoint from the student model and updating an ANN index of the corpus representation used for negative sampling. The encodings of the documents in the corpus can be updated multiple times during training, allowing us to sample new hard negatives from the ANN index more than once. In our reproducibility work, we update the ANN index only once. However, we also experiment with updating the ANN index multiple times (i.e., every 75k steps). In our preliminary experiments, updating and sampling from the ANN index more than once does not significantly increase performance: MRR@10 = .374 for MS MARCO, NDCG@10 = .714 for TREC-DL'19, and NDCG@10 = .685 for TREC-DL'20 (these scores can be compared against the first row in Table 4). We believe this can be attributed to the availability of high-quality hard negatives from the first update itself, since the student model is initialized with TAS-B.

**Training strategy.** CLDRD, which is the primary competing approach of MRL, uses a different training setup when training with listwise knowledge distillation. In particular, CLDRD constructs the training batch using the teacher model to produce the data, the teacher's relevance judgments have the form of pseudo-labels (w.r.t. controlling term $\mathbb{1}\{y_q^t(d) > y_q^t(d')\}$ in Eq. 12), it does not use in-batch negatives, and it uses curriculum learning (see Section 4.2 and Appendix C for more details). In contrast, MRL constructs the training batch via ground-truth query relevance judgments (i.e., qrel file) and negative mining, and the teacher's relevance judgments are the raw scores from the model. We proceed with using CLDRD's training setup to train MRL; we refer to this model variant as MRL-CLDRD. This approach ensures a fair comparison with CLDRD and allows us to attribute any performance increase solely to MRL's multivariate representations. When we compare rows 1 and 5 in Table 4, we observe that MRL can slightly benefit from incorporating the training setup of CLDRD. Furthermore, by comparing rows 5 and 6, we notice that the multivariate representations cause a slight decrease in performance, consistent with the previous result.

Table 5: Retrieval results for MRL when trained with 512 batch size and for different numbers of training steps.

| Steps | Training time (hours) | MS MARCO MRR@10 | TREC-DL'19 NDCG@10 | TREC-DL'20 NDCG@10 | SciFact NDCG@10 | FiQA NDCG@10 | TREC-COVID NDCG@10 | CQADupStack NDCG@10 |
|---|---|---|---|---|---|---|---|---|
| 5880 | 79 | .312 | .680 | .641 | .365 | .226 | .407 | .278 |
| 11760 | 162 | .330 | .693 | .656 | .517 | .249 | .457 | .293 |

## 5.4 The Effect of Batch Size on MRL Performance

In this section, we explore whether increasing the batch size can achieve the reported retrieval results of the original paper, with the following goals: (i) we study the impact of batch size on the performance of MRL, and (ii) we unveil to what extent the reported results of the original work are due to the use of large batch size. Note that the original study uses a batch size of 512. In contrast, we use a batch size of 15, owing to our memory-limited, single GPU setup (i.e., an academic computation budget). In our implementation, a batch size of 512 would require around 34 A100 GPUs, which is unfeasible given our academic computation budget.

We adopt a memory reduction, gradient caching method to resolve this limitation. Introduced by Gao et al. (2021), GradCache allows us to train models using a batch-wise contrastive loss with a large batch on a single GPU[1]. The original study empirically demonstrates that GradCache can reproduce state-of-the-art results of a dense retriever – originally trained on multiple GPUs with a large batch size – on a single consumer-grade GPU, albeit with a significant increase in training time. We experiment with the following batch sizes: 15, 32, 64, 128, 256, and 512. Since there is significant additional training time due to GradCache's computational overhead, we train all models for the same number of epochs. We experiment with MRL trained with in-batch negatives (as proposed in the original paper) and without in-batch negatives.

As Figures 4 and 5 show, changes in batch size have a smaller impact on MRL when trained without in-batch negatives rather than with in-batch negatives. We hypothesize that the negative trend as the batch size increases is linked to an increase in the number of <hard-negative, easy-negative> passage pairs contributing to the loss – where hard-negatives are passages sampled from BM25 and the student model while easy-negatives are the in-batch negatives. As the batch size increases, due to the use of in-batch negatives, there is an increase in <positive, easy-negative> passage pairs and an even greater increase in <hard-negative, easy-negative> pairs. The latter demands the dense retriever to recover finer-grained distinctions between the passages. Therefore, it is possible that as the batch increases, the number of such pairs also increases; consequently, the training steps should also increase so that the model can learn these fine-grained differences.

To validate this, we train MRL with a batch size of 512 for more training steps to examine whether that could lead to better performance. As we can see in Table 5, training for more steps can increase retrieval performance on all datasets. Thus, we hypothesize that the reported results in the original paper might be an outcome of training with 512 for more training steps. Testing this hypothesis on our memory constraint environment, which consists of a single GPU and employs GradCache to simulate training with a large batch, is not feasible; as we show in Table 5, training MRL with batch size 512 on a single GPU with GradCache requires ∼8 days for 11K steps. At this point, we need to underline that such a result would suggest that the trends presented in the original paper may have been influenced by the fact that MRL was training with a significantly larger batch size compared to its baselines (i.e., in the original work, MRL is trained with a 512 batch, while CLDRD with a batch size of 8).

## 5.5 A simple extension to reduce the hyperparameter search space

The MRL model produces a mean $(\mu_Q, \mu_D)$ and a diagonal co-variance matrix/variance vector $(\Sigma_D, \Sigma_Q \ / \ \sigma_Q, \sigma_D)$ given text input. Using the raw co-variance without ensuring that it is positive (and semi-definite)

---

[1]Even though we can avoid encoding extra negatives with in-batch negatives, the loss of each sample is conditioned on every sample in the batch and therefore it is required to fit the entire large batch into GPU memory

Table 6: MRL Variant: LogVar vs Softplus: LogVar performs similarly to Softplus, but with no hyperparameter tuning required. Based on a two-tailed paired t-test ($p < 0.05$), there are no statistically significant differences between the models results.

| Variant | MS MARCO MRR@10 | TREC-DL'19 NDCG@10 | TREC-DL'20 NDCG@10 | SciFact NDCG@10 | FiQA NDCG@10 | TREC-COVID NDCG@10 | CQADupStack NDCG@10 |
|---------|---------|------------|------------|---------|------|------------|-------------|
| Softplus | .373 | .707 | .681 | .591 | .291 | .497 | .327 |
| LogVar | .373 | .709 | .679 | .590 | .296 | .471 | .327 |

would make the model produce invalid values for the variance e.g., a negative variance. To ensure positivity, Zamani & Bendersky (2023) pass the raw values through a *softplus* activation function. If $a$ is the output of the encoder (i.e., prior to passing through the *softplus*), then the variance is obtained as follows:

$$\sigma = \text{Softplus}(a) = \frac{1}{\beta} \cdot \log(1 + \exp(\beta \cdot a)), \tag{13}$$

which ensures that the predicted variance is positive. The original study finds that retrieval performance is robust to the hyperparameter $\beta$ in the *softplus* function. An alternative to using the *softplus* function (that predicts a variance), is to assume that the output $a$ is the *log*-variance, without passing it through any activation function. Therefore, Equations 1 and 2 become:

$$(\mu_Q, \log \Sigma_Q) = f_\theta(q), \tag{14}$$

$$(\mu_D, \log \Sigma_D) = f_\phi(d). \tag{15}$$

As such, these values are exponentiated element-wise prior to use e.g., when plugging it into Eq. 11 and Eq. 10. This alternate method renders tuning the $\beta$ hyperparameter unnecessary. We term this variant "LogVar", and validate this approach through experimentation.

This experiment compares the MRL model trained using the *softplus* activation with the LogVar variant. We stress that we do not expect improved performance; this is a variant that functions similarly to the original model. The results, reported in Table 6, show that the LogVar model achieves similar performance across datasets and metrics. We also conducted an ablation with different initializations as well as without distillation and observed the same trend. In essence, the LogVar model provides results similar to those of the Softplus model without requiring an additional hyperparameter.

### 5.6 Limitations of our study

**Lack of details.** As mentioned previously, our reproducibility effort was hampered by a lack of crucial details in the original paper, such as the composition of the batches used while training the model, or which underlying cross-encoder was used as a teacher. Therefore, it is unclear whether the impressive performance improvements reported in the original paper are the result of some crucial implementation detail that was omitted. However, we tried to mitigate this limitation through *exhaustive* experimentation. The most promising part of that experimentation is presented in our ablation study, but it still constitutes a fraction of all model configurations that we attempted in order to get closer to the reported performance.

**Batch size.** As mentioned in Section 5.4, we rely on GradCache (Gao et al., 2021) to support training MRL with larger batches. As a result, training time when training with a larger batch size does not decrease training time since we are still limited to a single GPU. In contrast, training time increases significantly due to GradCache's overhead of calculating and storing gradients of representations. Due to this limitation, we cannot fully explore training MRL with a batch of 512 for longer (i.e., more training steps).

## 6 Conclusion

In this paper, we reproduce the multivariate representation learning framework by Zamani & Bendersky (2023) in a memory-constrained environment. After addressing a likely typographical error in the original

paper's derivations, we show that in a fair comparison, MRL achieves similar performance to baseline models. Crucially, the main claim - that MRL significantly outperforms baseline models in out-of-domain datasets - does not hold in our experimental setup. While MRL does not outperform baselines, we maintain that it remains a competitive retrieval model. We also conduct an extensive analysis of the predicted variance. Against our expectations, our analysis reveals that the variance vectors do not consistently express uncertainty. To add to the results of the original study, we conduct a thorough ablation study, investigating the impact of the different components of the MRL framework: (i) multivariate representations, (ii) distillation, and (iii) model initialization. Through this study, we conclude that multivariate representations do not improve or harm performance significantly, and knowledge distillation is the primary source of improvement. In addition, we show that models trained with an increased batch size using gradient caching methods and without increasing the number of training epochs are impacted negatively by the presence of in-batch negatives.

While we are unable to reproduce the results, we maintain that the ideas in the original paper are very valuable to the community. Prior to this paper, uncertainty was only utilized in ranking, not first-stage retrieval. The decomposition of the KL divergence as a dot-product enables incorporating uncertainty in first-stage retrieval. This implies that *any* model that produces a distribution for queries/passages can be used in this framework, even if it was not trained with the objective function outlined in the paper – this is a promising direction for future research. Future work could further consider incorporating *document* uncertainty to the framework, e.g., for post-retrieval QPP.

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

## A  Dot product formulation in the original paper

We include the original dot product formulation below of the KL divergence for completeness. Consider the term $\text{tr}\{\Sigma_D^{-1}\Sigma_Q\}$, the source of the error. Zamani & Bendersky (2023) formulate this as:

$$\text{tr}\{\Sigma_D^{-1}\Sigma_Q\} = \frac{\prod_{i=1}^k \sigma_{i_Q}^2}{\prod_{i=1}^k \sigma_{i_D}^2}. \tag{16}$$

This differs from our version in Eq. 6. This error is propagated throughout the next steps. The KL formulation becomes:

$$\text{KLD}(Q\|D) = \sum_{i=1}^k \log\sigma_{i_D}^2 + \frac{\prod_{i=1}^k \sigma_{i_Q}^2}{\prod_{i=1}^k \sigma_{i_D}^2} + \sum_{i=1}^k \frac{\mu_{i_Q}^2}{\sigma_{i_D}^2} - \sum_{i=1}^k \frac{2\mu_{i_Q}\mu_{i_D}}{\sigma_{i_D}^2} + \sum_{i=1}^k \frac{\mu_{i_D}^2}{\sigma_{i_D}^2}. \tag{17}$$

The equivalent dot product formulation, after taking the negative of the equation above gives us the original formulation (Eq. (14) in Zamani & Bendersky (2023), with signs flipped for $\vec{d}$):

$$\vec{q} = \left[1, \prod_{i=1}^k \sigma_{i_Q}^2, \mu_{1_Q}^2, \ldots, \mu_{k_Q}^2, \mu_{1_Q}, \ldots, \mu_{k_Q}\right], \tag{18}$$

$$\vec{d} = \left[\gamma_D, \frac{1}{\prod_{i=1}^k \sigma_{i_D}^2}, \frac{1}{\sigma_{1_D}^2}, \ldots, \frac{1}{\sigma_{k_D}^2}, \frac{2\mu_{1_D}}{\sigma_{1_D}^2}, \ldots, \frac{2\mu_{k_D}}{\sigma_{k_D}^2}\right], \tag{19}$$

where $\gamma_D$ is equivalent in our formulation i.e., Eq. 9. Here, $\vec{q}, \vec{d} \in \mathbb{R}^{1\times(2k+1)}$ and $q^\intercal \cdot d$ is equal to Eq. 17. Note that the final vector for $\vec{d}$, the signs are flipped because the similarity function is the *negative* KL divergence.

## B  Dot product formulation of full KL divergence

We start from the *unsimplified* version of Eq. 8, that includes constants:

$$\text{KLD}(Q\|D) = \frac{1}{2}\left[\sum_{i=1}^k \log\sigma_{i_D}^2 - \sum_{i=1}^k \log\sigma_{i_Q}^2 - k + \sum_{i=1}^k \frac{\sigma_{i_Q}^2}{\sigma_{i_D}^2} + \sum_{i=1}^k \frac{\mu_{i_Q}^2}{\sigma_{i_D}^2} - \sum_{i=1}^k \frac{2\mu_{i_Q}\mu_{i_D}}{\sigma_{i_D}^2} + \sum_{i=1}^k \frac{\mu_{i_D}^2}{\sigma_{i_D}^2}\right]. \tag{20}$$

To formulate it as a dot product, we use the definition of the document prior, $\gamma_D$, from Eq. 9, but further define $\gamma_Q$ as:

$$\gamma_Q = \sum_{i=1}^k \log\sigma_{i_Q}^2. \tag{21}$$

Then we can extend the vector representations in Eq. 10 and Eq. 11 to include $\gamma_Q$:

$$\vec{q'} = \left[1, \gamma_Q, \sigma_{1_Q}^2, \ldots, \sigma_{k_Q}^2, \mu_{1_Q}^2, \ldots, \mu_{k_Q}^2, \mu_{1_Q}, \ldots, \mu_{k_Q}\right], \tag{22}$$

$$\vec{d'} = \left[\gamma_D, 1, \frac{1}{\sigma_{1_D}^2}, \ldots, \frac{1}{\sigma_{k_D}^2}, \frac{1}{\sigma_{1_D}^2}, \ldots, \frac{1}{\sigma_{k_D}^2}, -\frac{2\mu_{1_D}}{\sigma_{1_D}^2}, \ldots, -\frac{2\mu_{k_D}}{\sigma_{k_D}^2}\right], \tag{23}$$

where $\vec{q'}, \vec{d'} \in \mathbb{R}^{1\times(3k+2)}$. Then,

$$\text{KLD}(Q\|D) = \frac{1}{2}\left(\vec{q'}^\intercal \cdot \vec{d'} - k\right), \tag{24}$$

should precisely yield the KL divergence between the distributions of $Q$ and $D$, as defined in Eq. 4.

Table 7: Statistics and Description of Evaluation Datasets. Number of tokens for average query/document lengths were computed based on the `distilbert-base-uncased` tokenizer.

| | Name | Domain | # q | # p | avg. q length | avg. p length |
|---|---|---|---|---|---|---|
| **ID** | MS MARCO Dev | Miscellaneous | 6,890 | 8,841,823 | 9.01 | 76.97 |
| | TREC-DL 19 | Miscellaneous | 43 | 8,841,823 | 9.02 | 76.97 |
| | TREC-DL 20 | Miscellaneous | 54 | 8,841,823 | 9.22 | 76.97 |
| **OOD** | Scifact | Scientific Document Retrieval | 300 | 5,183 | 22.84 | 315.65 |
| | FiQA | Financial QA | 648 | 57,638 | 15.59 | 177.11 |
| | TREC-COVID | Biomedical document retrieval | 50 | 171,332 | 18.04 | 224.78 |
| | CQADupStack | Community QA retrieval | 13,145 | 457,199 | 13.55 | 248.73 |

## C  Training setup for CLDRD

One of the perks of training with knowledge distillation is that it can leverage the incomplete relevance judgments that most large-scale retrieval datasets suffer from. Hence, it can rely only on the teacher supervision signal, i.e., training data is generated and scored by the teacher model without human assessments. Furthermore, combining knowledge distillation with in-batch negative training can be impractical when the teacher is a computationally expensive cross-encoder (Lin et al., 2021). On the other end of the spectrum, even though the exact relevance scores produced by the teacher model do not impact the loss value (since the loss only considers the order of passages) they still play an important role in controlling which query-passage pairs will contribute to the loss (term $\mathbb{1}\{y_q^t(d) > y_q^t(d')\}$ in the loss). When the raw scores from the teacher model are used, all pairs contribute to the loss, even when contrasting two irrelevant passages.

In contrast to the training setup used by Zamani & Bendersky (2023) (see Section 3.2), the work by Zeng et al. (2022)–the work that initially proposed the listwise distillation loss used by Zamani & Bendersky (2023) and the primary competing approach of MRL–uses pseudo-labeling to create the batch and does not rely on in-batch negatives. The authors use the following approach to compute the listwise distillation loss:

- Given a query $q$, the passage set $D_q$ is constructed with respect to the top-k passages in the ranked list returned by the teacher model (reranking order). In particular, the first $K$ passages in the ranked list returned by the teacher model are considered positive, the next $K'$ are considered hard negatives, and the remaining $K''$ soft negatives.

- $y_q^t(d)$ is a relevance label according to the group, passage $d$ belongs to:

$$y_q^t(d) = \begin{cases} \frac{1}{r_{qd}^t} & \text{iff d is positive} \\ 0 & \text{iff d is hard-negative} \\ -1 & \text{iff d is soft-negative} \end{cases}$$

where $r_{qd}^t$ is the ranking position of the document $d$ given the query $q$ in the teacher ranked list.

## D  Datasets

In Table 7 we present the statistics of the datasets. Here, "p" and "q" indicate questions and passages, respectively. The length is in tokens.

## E  Hyperparameters

In Table 8 we detail our hyperparameter search space used throughout our experimental setup. For MRL we search w.r.t. learning rate, $\beta$, and negatives. The best parameters were: $\beta = 2.5$, $lr = 5 \times 10^{-6}$, 5 negative passages from BM25, and 25 negative passages from the student model. For MRL-CLDRD we search w.r.t.

Table 8: Hyperparameter search space for the models we experiment with. Negatives are presented in the following format:$[\#ANN\ negatives, \#BM25\ negatives]$

| Parameter | Values |
|---|---|
| Learning rate | $1 \times 10^{-4}$, $1 \times 10^{-5}$, $1 \times 10^{-6}$, $3 \times 10^{-6}$, $5 \times 10^{-6}$, $7 \times 10^{-6}$ |
| $\beta$ | 0.5, 1, 2.5, 7.5 |
| Negatives | $[5, 25]$, $[10, 20]$, $[15, 15]$, $[20, 10]$, $[25, 5]$ |

learning rate and $\beta$. The best parameters were: $\beta = 2.5$, and $lr = [5 \times 10^{-6}, 1 \times 10^{-6}, 1 \times 10^{-6}]$ for the three curriculum iterations. For CLDRD the best learning rates for each of the three curriculum iterations were $[7 \times 10^{-6}, 3 \times 10^{-6}, 3 \times 10^{-6}]$, while for DPR the best learning rate was $7 \times 10^{-6}$.

Our academic computational budget allowed us to perform a hyperparameter search only on the models in Table 1 and the model using cross-entropy without distillation in Table 4. However, we note that the model performance is quite robust to the choice of hyperparameters in our initial experiments – mitigating this limitation to an extent.

