# OpenReview forum: "Multivariate Dense Retrieval: A Reproducibility Study under a Memory-limited Setup"
_TMLR — Accepted by TMLR_

### Review · Reviewer_FS2P · 2024-10-10

**Summary Of Contributions:**

This paper's contributions are as follows:
1. The authors identified and corrected a key mathematical error in the original Multivariate Representation Learning (MRL) framework. This error occurred early in the formulation of KL divergence and may have propagated throughout the model’s mathematical foundations. The correction leads to more accurate retrieval results compared to the original formulation in the original work.

2. The authors attempted to reproduce the MRL results using a memory-limited setup (1xA100 40GB). They found that the results could not be reproduced under these constrained settings, demonstrating that the performance of MRL heavily depends on large batch sizes and extended training times.

3. Through an ablation study (Sec 5.3), the authors concluded that the performance gains attributed to the original MRL paper are largely due to knowledge distillation and model initialization, rather than the multivariate representations themselves. The multivariate aspects of MRL, which were intended to model uncertainty, did not significantly contribute to performance improvements.

4. The analysis (Sec 5.2.2) showed that the variance component of MRL **does not capture uncertainty**, as initially proposed. For example, in Figure 3, the model assigns higher uncertainty to clean queries rather than noisy ones, contradicting the assumptions of the MRL paper.

5. The authors propose an improvement to the MRL framework that reduces the hyperparameter search space by replacing the Softplus activation with a log-variance. This adjustment maintains competitive performance without the need for tuning specific parameters, thus enhancing the practicality of the model.

**Audience:**

Yes

**Claims And Evidence:**

Yes

**Requested Changes:**

Please see "Minor errors" in section above. I may not have captured all of these, so please have a careful read again to make sure the references to the updated Tables are correct.

**Strengths And Weaknesses:**

### Strengths
1. The authors have addressed the concerns of the previous reviewers in regard to the language used to describe the work of Zamani & Bendersky (2023). The tone of the paper has become more neutral and reserved. This is good, as there are many aspects of the reproducibility study that cannot be exactly the same as the original paper, as already discussed by the authors.
2. The authors have introduced a new experiment section discussing the impact of large batch sizes. The study indicates that the performance of MRL heavily depends on large batch sizes and extended training times.
3. The paper rectifies errors in Zamani & Bendersky (2023), likely to be unknown to the authors, that should be highlighted. While this is a potentially risky endeavour, as highlighted by reviewer iL1v, the impact will be a net-positive for the progress in the community.
4. The writing, presentation, and experiments are of very high quality.

### Weaknesses
1. The study does not fully replicate the results of Zamani & Bendersky (2023) due to resource constraints, notably the inability to complete experiments at batch size 512. Therefore, some of the results (e.g. Table 1) should be interpreted with caution and the conclusions are left open-ended.
2. There are errors in the discussion when referring to Table 1 in Section 5.1. See details below.

### Minor errors
1. Sec. 5.1 (page 9): "Regarding MRL…. For instance, in the case of FiQA, our implementation yielded an NDCG@10 of **0.308**".
- Table 1 shows **0.293**, not 0.308 for MRL.
2. Sec. 5.1 (page 9): "A similar trend holds for the OOD dataset, except for TREC-COVID, where CLDRD outperforms MRL with a substantially higher score of 0.608 compared to **0.481** for MRL."
- Table 1 shows **0.497**, not 0.481 for MRL.

---

> ### Author Response · Authors · 2024-10-14
> **Response to reviewer FS2P**
>
> Thank you for your valuable feedback. We greatly appreciate your attention to detail, particularly in identifying the typos mentioned in the "Minor errors" section. We will ensure that all these corrections are made. Furthermore, we will carefully review the entire manuscript again to catch any additional typographical errors.

---

### Review · Reviewer_LzJ1 · 2024-11-10

**Summary Of Contributions:**

This paper aims to reproduce multivariate representation learning (MRL) under memory constraints. It has spotted an error in the original MRL paper and reproduced some results in a memory-limited setting, even though the original paper has not clearly specified some key algorithm parameters. This paper has also done a thorough ablation study, and open-sourced their code.

This paper has confirmed that MRL can have the state-of-the-art performance.

**Audience:**

Yes

**Broader Impact Concerns:**

None.

**Claims And Evidence:**

Yes

**Requested Changes:**

- Please explain the novelty of this paper.

**Strengths And Weaknesses:**

**Strengths:**

- very well written and easy to follow

- rigorous and extensive empirical analyses of MRL

**Weaknesses:**

- this paper aims to reproduce results from a previous paper. Thus, it seems that this paper lacks novelty.

---

> ### Author Response · Authors · 2024-11-12
> **Response to reviewer LzJ1**
>
> Thank you for your feedback.
>
> This is indeed, a reproducability study. There has been considerable research in several fields showing that a significant portion of research is irreproducible [1, 3, 4], making reproducibility studies an important way to test the validity of ideas and methods. Top-tier journals and conferences have incorporated reproducibility challenges, tracks, and workshops to emphasize the importance of such efforts (e.g., the  SIGIR reproducibility track: https://sigir-2024.github.io/call_for_res_rep_papers.html and the TMLR reproducibility challenge: https://reproml.org/). Furthermore, TMLR itself accepts reproducibility studies of works that have not been reproduced before:  https://jmlr.org/tmlr/editorial-policies.html.
>
>
> In our work, there are a number of new insights and findings that are not present in the original paper. We state them explicitly in the introduction section and throughout the paper. In summary:
>
> 1. We correct a typographical/mathematical error crucial for reproducing the original work. We showed that it is impossible to obtain competitive retrieval performance without correcting this error.
>
> 2. Reproducibility is primarily concerned with generalization to different settings [2, 3]. In our work, we test the effectiveness of the original framework under a memory-limited setup,  in contrast to the original work, which assumes the availability of industry-grade computational resources.
>
> 3. We showcase the inability of the original framework to capture uncertainty under a memory-limited setup. To this extent, regarding investigating the predicted variance, we further introduce two different sets of experiments (beyond the QPP that was used in the original paper): (a) contrasting the predicted variance of corrupted and clean data and (b) experimenting with alternate encoding schemes.
>
> 4. We provide an extensive ablation study on the model’s components to unveil their importance in training an effective retriever. Such ablation was not present in the original paper. Most importantly the competitive retrieval performance of MRL is not a result of using multivariate representations.
>
> 5. We propose the use of LogVar. The main intent was to use an alternative to ensure that the co-variance remains positive semi-definite. One way to ensure positivity is to use a softplus activation, which is the approach taken in the original paper. Predicting the log-variance instead of using the softplus – and exponentiating the log-variance when needed – would also ensure positivity, with the advantage of not needing a hyperparameter search for $\beta$.
>
> 6. We are releasing our code and providing model checkpoints to enhance reproducibility and facilitate further exploration.
>
> [1] Baker. 2016. Is there a reproducibility crisis? Nature.
>
> [2] ACM. 2018. Artifact Review and Badging.
>
> [3] Pineau et al. 2020. Improving Reproducibility in ML Research
>
> [4] Hutson. 2018. Artificial intelligence faces reproducibility crisis.

---

> > ### Comment · Action_Editor_qkTS · 2024-11-21
> > **Comments on the rebuttal**
> >
> > Dear Reviewer LzJ1,
> >
> > Thank you for your efforts in reviewing the paper. The authors have provided a description of the novelty of the approach as required. In particular, beyond pointing out some errors in [1], one of the contributions of the paper is to demonstrate that the covariance matrix doesn’t correlate with uncertainty, which introduces a debate as to the relationship between the intuition of probabilistic methods and the true mechanism through which they work.  There is also an additional methodological component, **LogVar**.
> >
> > I would be keen to hear your thoughts on all these aspects.
> >
> >
> >
> > [1] Hamed Zamani, Michael Bendersky. “Multivariate Representation Learning for Information Retrieval “. SIGIR 2023.

---

> > ### Comment · Reviewer_LzJ1 · 2024-12-29
> > **Thanks!**
> >
> > Thanks a lot for the detailed responses! They have addressed my concerns.

---

### Review · Reviewer_A5fF · 2024-11-25

**Summary Of Contributions:**

The paper thoroughly investigates the reproducibility and performance of the multivariate representation learning (MRL) framework proposed by Zamani & Bendersky (2023) for dense retrieval tasks.

**Audience:**

Yes

**Broader Impact Concerns:**

There is no concern about the ethical implications of this work.

**Claims And Evidence:**

Yes

**Requested Changes:**

1) Conduct a thorough analysis of the MRL framework's scalability. Provide a detailed discussion on scalability, including potential bottlenecks and strategies for optimization, to address the unclear scalability issue.
2) Updated Comparison Method. Include comparisons with the latest state-of-the-art methods from 2022 and beyond. Justify the choice of new comparison methods and discuss how they differ from the older ones, highlighting any advantages or disadvantages.

**Strengths And Weaknesses:**

Strengths:
1) The paper corrects a mathematical error in the original MRL formulation and makes reasonable design choices where the original paper was unspecified.
2) The paper provides a comprehensive ablation study that dissects the impact of different components of the MRL framework, offering valuable insights into what contributes to the effectiveness of the model.
3) The paper openly shares the source code for the reproducibility study.
Weaknesses:
1) While the paper attempts to address memory limitations, the scalability of the MRL framework to even larger datasets or more complex models is unclear. The framework's performance might degrade or become computationally infeasible as the scale increases.
2) Comparison methodology is slightly outdated, earlier than 2022, and it is suggested that it be compared with newer state-of-the-art methods.
3) Despite the authors' best efforts, they were unable to fully reproduce the results reported in the original paper. This could be due to omitted details or unaccounted-for factors in the original work.

---

> ### Author Response · Authors · 2024-11-27
> **Response to reviewer A5fF**
>
> Thank you for your feedback.
>
> **Concerning scalability.** In our work, we have explored scalability across the following dimensions:
>
> 1. Dataset size: We have investigated the scalability of MRL on datasets of varying sizes. In Table 7 in the Appendix section we report the scales of each dataset (we post the table below for your reference). If you find this information critical, we would be happy to integrate Table 7 into the main text for better accessibility.
>
> |     Name     |             Domain            | $\#$ q |   $\#$ p  | avg. q length | avg. p length |
> |:------------:|:-----------------------------:|:------:|:---------:|:-------------:|:-------------:|
> | MS MARCO Dev |         Miscellaneous         |  6,890 | 8,841,823 |      9.01     |     76.97     |
> |  TREC-DL 19  |         Miscellaneous         |   43   | 8,841,823 |      9.02     |     76.97     |
> |  TREC-DL 20  |         Miscellaneous         |   54   | 8,841,823 |      9.22     |     76.97     |
> |    Scifact   | Scientific Document Retrieval |   300  |   5,183   |     22.84     |     315.65    |
> |     FiQA     |          Financial QA         |   648  |   57,638  |     15.59     |     177.11    |
> |  TREC-COVID  | Biomedical document retrieval |   50   |  171,332  |     18.04     |     224.78    |
> |  CQADupStack |     Community QA retrieval    | 13,145 |  457,199  |     13.55     |     248.73    |
>
> 2. Batch size: In Section 5.4, we explore the effect of batch size in training MRL. Since we are limited to a single GPU, to train with larger batch sizes, we use a gradient caching technique, GradCache, to unlimitedly scale contrastive learning batches far beyond the GPU memory constraint [1].
>
>
> We think the reviewer's intended meaning is scalability with respect to the number of model parameters (e.g., replacing the BERT backbone, with a BERT-large backbone). We acknowledge that it is a compelling direction for future work, but we cannot see how it fits the "memory-limited" aspect of our reproducibility study. To better address your concerns, could you please clarify which specific scalability experiments you would like to see?
>
> **Concerning the baselines.** We would like to clarify the reasoning behind using these particular baselines. In detail:
>
> 1. Uncertainty in dense retrieval: First and foremost, MRL is up to this date the only work in dense retrieval that works in the direction of capturing uncertainty--this also underlines the significance of reproducing this work. Therefore, there is no other approach in dense retrieval that captures uncertainty and can be used as a direct competitor. While it is true that more recent dense retrieval methods surpass MRL in performance, these methods do not share the same focus on capturing uncertainty, making direct comparisons less relevant to our study's scope.
>
> 2. Alignment with reproducibility goals: Since this is a reproducibility study, we prioritized using the models used in the original work as baselines. This approach ensures a direct and fair verification of the claims made in the original paper.
>
> 3. Facilitating fair comparisons in ablation studies: Using these particular baselines facilitates fair comparisons in our subsequent ablation study.
>
> 	- MRL can be compared with CLDRD to assess the impact of the multivariate representations.
> 	- MRL can be compared with TASB to assess the impact of using the latter as an initialization point for the former.
> 	- A similar assessment can be made when MRL without distillation is compared against DPR.
>
> To conclude, we would happily add as a reference some of the latest works in dense retrieval if you really believe it can benefit the paper.
>
> References
> [1] Gao, Luyu, et al. "Scaling deep contrastive learning batch size under memory limited setup." arXiv preprint arXiv:2101.06983 (2021).

---

### Review · Reviewer_2tbi · 2024-12-06

**Summary Of Contributions:**

The paper rigorously walk through the Multivariate Representation Learning (MRL) framework and corrects the error in the existing work.

**Audience:**

Yes

**Broader Impact Concerns:**

No concerns regarding ethical implications.

**Claims And Evidence:**

Yes

**Requested Changes:**

See the weakness part.

**Strengths And Weaknesses:**

Strenghts:
1. The paper rigorously reviews the Multivariate Representation Learning (MRL) framework and corrects the error in the existing work, demonstrating that MRL does not outperform baselines, which contrasts the results they reported.
2. They further spot that multivariate representation itself is not the main source of the quality improvement, but the knowledge distillation is.

Weakness:
1. It is not the critical weakness since this paper is mainly about reproducing MRL, but it could improve more by including the reproduced results of more recent works [1].

[1] Zhou, Xixi, Gao, Yang, Jie, Xin, Cai, Xiaoxu, Bu, Jiajun, and Wang, Haishuai. "EASE-DR: Enhanced Sentence Embeddings for Dense Retrieval." In Proceedings of the 47th International ACM SIGIR Conference on Research and Development in Information Retrieval (SIGIR '24), Washington DC, USA, 2024, pp. 2374–2378. Association for Computing Machinery. doi:10.1145/3626772.3657925

---

### Author Response · Authors · 2024-12-07
**First Revision**

Dear reviewers and AE,

First and foremost, we would like to thank you for taking the time to review our work.

Since the reviewers are scheduled to begin their evaluation on December 9th, we are submitting an updated version of our manuscript. Please note that some of the revisions are based on our interpretation of the reviewers' expectations. If any of the changes fall short of meeting the reviewers' intentions, please do not hesitate to inform us, and we will make further revisions as necessary.

In the new version of our work, the changes are indicated with blue color. We trust that these updates will better address the reviewers’ concerns and enhance the clarity and quality of our work.

At this point, we would like also to clarify some things.

This is a reproducability work, and as it is commonly done it is focused on reproducing a single paper. In our work, we focus on reproducing "Multivariate Representation Learning for Information Retrieval" [4], namely, MRL. However, as part of our research, we further reproduced the paper "Curriculum Learning for Dense Retrieval Distillation" [5], namely, CLDRD. We believe that reproducing two papers in one study provides significant value and insight. To this end, we want to respectfully argue that reproducing more than two works in a single reproducibility paper should not be considered a mandatory requirement.

Moreover, we would like to stress once again what are some of the main contributions of our reproducibility work:
1. We correct a typographical/mathematical error crucial for reproducing the original work. We showed that it is impossible to obtain competitive retrieval performance without correcting this error.
2. Reproducibility is primarily concerned with generalization to different settings [2, 3]. In our work, we test the effectiveness of the original framework under a memory-limited setup, in contrast to the original work, which assumes the availability of industry-grade computational resources.
3. We showcase the inability of the original framework to capture uncertainty under a memory-limited setup. To this extent, regarding investigating the predicted variance, we further introduce two different sets of experiments (beyond the QPP that was used in the original paper): (a) contrasting the predicted variance of corrupted and clean data and (b) experimenting with alternate encoding schemes.
4. We provide an extensive ablation study on the model’s components to unveil their importance in training an effective retriever. Such ablation was not present in the original paper. Most importantly the competitive retrieval performance of MRL is not a result of using multivariate representations.
5. We propose the use of LogVar. The main intent was to use an alternative to ensure that the co-variance remains positive semi-definite. One way to ensure positivity is to use a softplus activation, which is the approach taken in the original paper. Predicting the log-variance instead of using the softplus – and exponentiating the log-variance when needed – would also ensure positivity, with the advantage of not needing a hyperparameter search for $\beta$.
6. We are releasing our code and providing model checkpoints to enhance reproducibility and facilitate further exploration.

Thank you for your understanding and continued support throughout this process.


[1] Baker. 2016. Is there a reproducibility crisis? Nature.

[2] ACM. 2018. Artifact Review and Badging.

[3] Pineau et al. 2020. Improving Reproducibility in ML Research

[4] Hutson. 2018. Artificial intelligence faces reproducibility crisis.

[5] Zamani and Bendersky. 2023. Multivariate representation learning for information retrieval.

[6] Zeng et al. 2022. Curriculum learning for dense retrieval distillation.

---

### Decision · Action_Editor_qkTS · 2024-12-28

**Recommendation:** Accept as is

**Comment:**

This paper attempts to reproduce the results of Multivariate Representation Learning (MRL)  [1] and identify and correct an issue in the mathematical derivations in the original paper which occurred in the expression of the KL divergence. In addition, they show that under a limited batch size setting, MRL doesn't consistently outperform the baselines. The authors of the present paper credibly indicate that reproducing [1] under the original setting (with a batch size of 512) would be prohibitively expensive in terms of compute time. Ablation studies are performed to evaluate most of the components of the original model and a striking conclusion is the fact that, as reviewer FS2P mentions,  *the variance component of MRL does not capture uncertainty* and at best merely acts as additional modelling parameters. The true gains are instead derived from the knowledge distillation and initialisation.


In addition, the authors introduce a novel training component, LogVar, which eliminates the need for the Softplus function in MLR variants by interpreting the model based $\Sigma$ as the component-wise logarithm of the variance. It is shown that Logvar is competitive with softplus despite not requiring an additional hyperparameter.

The paper makes a solid contribution, has greatly improved during the reviewing process, and clearly deserves to be accepted.





[1] Hamed Zamani, Michael Bendersky. “Multivariate Representation Learning for Information Retrieval “. SIGIR 2023.

**Audience:**

This is a very interesting topic which perfectly aligns with TMLR's audience.

**Claims And Evidence:**

After a very long reviewing process involving two submissions and multiple rounds of online revisions, I am reasonably convinced the paper has reached a high level of rigour and quality. The mathematical description of the inaccuracies in the original paper are correct and have been duly softened in terms of tone.